# Variability in tactical behavior of futsal teams from different categories

**Murilo José de Oliveira Bueno** [1⊙]*, **Fabio Giuliano Caetano** [1‡], **Nicolau Melo de Souza** [1‡], **Sergio Augusto Cunha** [2‡], **Felipe Arruda Moura** [1⊙]

**1** Laboratory of Applied Biomechanics, Sport Sciences Department, State University of Londrina, Londrina, Brazil, **2** College of Physical Education, University of Campinas, Campinas, Brazil

⊙ These authors contributed equally to this work.
‡ These authors also contributed equally to this work.
* murilojobueno@gmail.com

**Data Availability Statement:** All relevant data are available at https://figshare.com/articles/Variability_of_tactical_behavior_of_futsal_teams_from_different_categories_/9772559/1 or https://doi.org/10.6084/m9.figshare.9772559.v1.

## Abstract

The aim of the present study was to analyze the time series of team spread during futsal official matches in the frequency domain for different categories. Using an automatic tracking method, trajectories of 258 players (excluding goalkeepers) were obtained, composed of 79 players from the under-15 (U15) category, 86 from the under-18 (U18), and 93 from the professional. We calculated the team spread defined as the Euclidean norm of the distance-between-player vector as a function of time. We applied the Fast Fourier Transformation method and calculated the median frequency for each time series of spread. The results of mean ± SD of the median frequency of the time series of spread from the first to the second half were significantly different only for the U15 category (first half, 1.04 ± 0.46, second half 1.40 ± 0.34). The frequency values differed significantly between the categories. The younger categories presented a higher frequency of spread oscillation than the professional category, which reflects the dynamics of the game between attack and defense sequences. The results provide insights into the features of the sport and present a basis for appropriate training interventions for players in each category, planning for future transition to the following category.

## Introduction

The futsal like other team sports have interpersonal interactions between teammates and opponent players. The teammates interact cooperating with each other on tactical strategies to achieve offensive and defensive success [1]. One of the objectives in the analysis of a sport is to develop methodologies to understand the physical, technical, and tactical characteristics during official matches and training sessions [2], aiding comprehension about the tactical actions performed by the players, interaction with the opponents, or recommendations established by the coaches during a match or training [3,4]. The organization of the players on the court may reflect the strategic actions and information about these tactical behaviors can be obtained by the quantification of the team surface area and spread [2].

**Funding:** The authors would like to thank CNPq (protocol number 446548/2014-6) and Fundação Araucária (protocol number 25372/2013) for the financial support for this research. This study was financed in part by the Coordenação de Aperfeiçoamento de Pessoal de Nível Superior - Brasil (CAPES) - Finance Code 001. São Paulo Research Foundation (FAPESP) grant# 2016/50250-1 -2017/20945-0 - 2018/19007-9. The funders had no role in study design, data collection and analysis, decision to publish, or preparation of the manuscript.

A new technique used for tactical analysis in team sports is to understand the variability of time series in the frequency domain, related to the distribution of players in the field during a soccer match (such as team surface area, spread). For any signal represented by a time series, it is possible to perform spectral analysis, showing the distribution of the fluctuation intensity of a signal in the frequency domain [5,6]. Variability in the time series of spread (a measure of distances between teammates) during a match reflects the speed with which players increase and decrease team compactness during attacking and defensive activities during a match [5]. For soccer, a previous study [5] reported a decrease in the frequency of this signal from the first to the second half of the match. This decrease may be related to the physical performance of the players during the match, since the distances covered and percentage of distances traveled at medium and high speed also decrease during matches in both football and futsal [7–10]. Additionally, oscillation seems to correspond to the frequency with which the teams exchange ball possession [5].

In recent years, studies have been carried out to understand the tactical organization of futsal players, interpersonal coordination, and spatial-temporal dynamics in official matches or in training sessions [11–15], as well as studies that try to understand if tactical variables can present differences according to different categories in futsal [16,17]. Generally, these studies focused on variables that provide information about relative space and dispersion covered by the futsal players on the court. However, tactical performance is also dependent on how fast the player and teammates are able to organize themselves on the court when ball possession exchange occurs [5]. Additionally, it is possible that younger players, due to their lower tactical capacity [18], will not be able to maintain possession of the ball for long periods, which may showed a higher variability of the team spread, result in lower proficiency to maintain possession of the ball. Thus, the variability analysis of the spread of futsal teams from different age groups allows characterization of the profile of the game dynamics for categories that are in the process of learning of the sport modality as well as professional category.

The aim of the present study was to analyze the time series of team spread during futsal official matches in the frequency domain for different categories. This study presents two hypotheses: a) the younger the category, the greater the oscillation frequencies of the time series of spread would be, and b) these frequencies would decrease from the first to the second period, regardless of the category.

## Materials and methods

### Participants and match sample

The Research Ethics Committee of the University approved this study (process number. 22514). For this study, the Ethics Committee considered that an informed consent form signed by the participants or guardians was not necessary. The study was conducted between 2014 and 2016.

We registered images of fifteen matches for the present study, of these 5 matches for each category. The U15 (n = 79 players, regional championship), U18 (n = 86 players, state championship) and professional category (n = 93 players, professional championship) were the groups analysis. All matches were recorded in the State of Paraná –Brazil during the qualifying phase of the championships for all categories (Maximum score difference for each category: U15 = 5 goals; U18 = 3 goals; Professional = 5 goals). The recording method employed up to three cameras at a sampling frequency of 30 Hz, fixed in high points of the gyms, each one covering up to one third of the court, with regions of overlap between them. After the matches, we transferred the images to computers and synchronized the cameras by identifying common events that occurred in these regions of overlap, such as the exact moment of a kick.

## Data collection

Was used an automatic tracking system to obtain the trajectory of each player via DVideo software [19,20]. For each match, we obtained a set of specific points, associated with the actual coordinate system of the court, and also determined the corresponding projection of these points in the image in DVideo *software*. The homography parameters of the image-object transformation were calculated based on the DLT (Direct Linear Transformation), providing the 2D coordinates of each player associated with the coordinate system of the court. The average error for the determination of player position was 0.098 m, and the average error for the distance covered was 0.8% [9].

Each player, from each team was numbered as $p$ = 1, 2, . . ., 12. Thus, we defined the two-dimensional coordinates of the players as p ($Xp(t)$, $Yp(t)$), where $t$ represents each time instant (in seconds). Next, we used a 3rd order low pass Butterworth digital filter with a cutoff frequency of 0.4 Hz to smooth the 2D coordinate data of all players. Having obtained the smoothed trajectories of all players in each match, we calculated team spread as a function of time. Additionally, we identified which teams had and did not have possession of the ball according to the criteria defined by Moura et al, [5]

## Data analysis

For each time instant $t$, we calculated the Euclidean distances of each player and their teammates. The distances between players were organized in symmetric matrix $D$ of order $m \, x \, n$, where $m$ = number of distance values between the players of the same team and $n$ = corresponding frames to each time instant $t$ [5,16]. The Euclidian Norm of each vector of the matrix $D$ was then calculated, corresponding to spread values at each $t$, according to the equation:

$$||\text{D}|| = \sqrt{\sum\nolimits_{j=1}^{p} |d_{nij}(t)|^2}$$

Where $p$ represents the number of players on the court of the same team and $d_{nij}$ represents the values of Euclidian distance between each pair of players from the same team. Larger $||D||$ values mean that players are more spread across the court. In contrast, lower values indicate that players present a more compact structure [5,16].

For any time series, it is possible to identify the distribution of the fluctuation intensities of the signals in the frequency domain using spectral analysis [6]. We used FFT (Fast Fourier Transformation) in the Matlab® environment to calculate the power distribution in the frequency domain of the spread and the ball possession time series. Subsequently, we identified the median frequency, defined as the frequency dividing the integral of the power spectrum into two equal parts. In order to facilitate understanding of the results, the unit of measure adopted for the frequency was cycles.min$^{-1}$.

## Statistical analysis

Prior to each analysis, we performed a Levene's variance test to verify the homoscedasticity of the data. As not all the tests met the assumptions, we performed a *Box-Cox* transformation to reduce data anomalies and heteroscedasticity. We used a two-way analysis of variance to verify if there were differences between the median frequencies of the time series of spread from the first to the second half (factor 1) and to verify if there were differences between the U15, U18, and professional categories (factor 2). In both analyzes, when differences were found, we performed a Tukey post-hoc test as a significant difference criterion. The significance level for all

analyses was $p<0.05$. Data are expressed as mean and standard deviation. The effect size (EF) for the variance analysis was calculated according to Cohen's $f$ [21].

Specifically, we analyzed the time series associated with possession of the ball for each match in a descriptive way, calculating the percentages of changes in the frequencies, from the first to the second half in all the matches.

## Results

Fig 1 presents examples of the time series of spread of the teams in the U15 (A), U18 (B), and professional (C) categories. Visual analysis of the time series between the teams demonstrated more oscillations of the time series of spread in the younger categories.

Fig 2 exemplifies the results of spectral analysis of the first and second halves for the U15 (A), U18 (B), and professional (C) categories. These results represent the frequencies of the oscillations of the time series spread.

Table 1 presents the values of the median frequencies of the time series of spread of each team in the first and second halves for each category. Fig 3 presents the mean and standard deviation values for the median frequencies of spread found for each category in the first and second halves. Statistical tests revealed that the median frequency differed between all

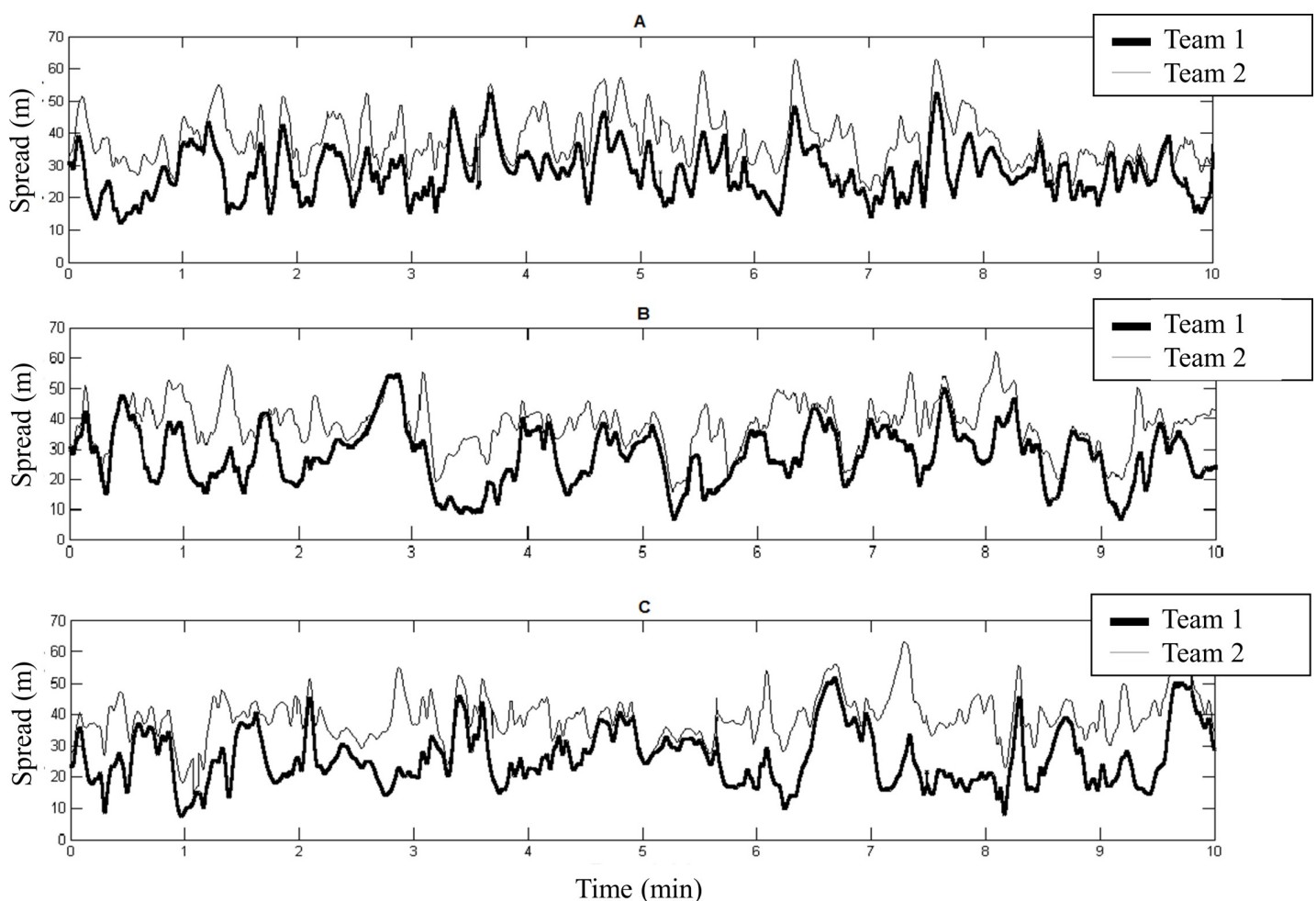

**Fig 1.** Time series of spread of different teams analyzed for the U15 (A), U18 (B), and professional (C) categories during ten minutes of the match.

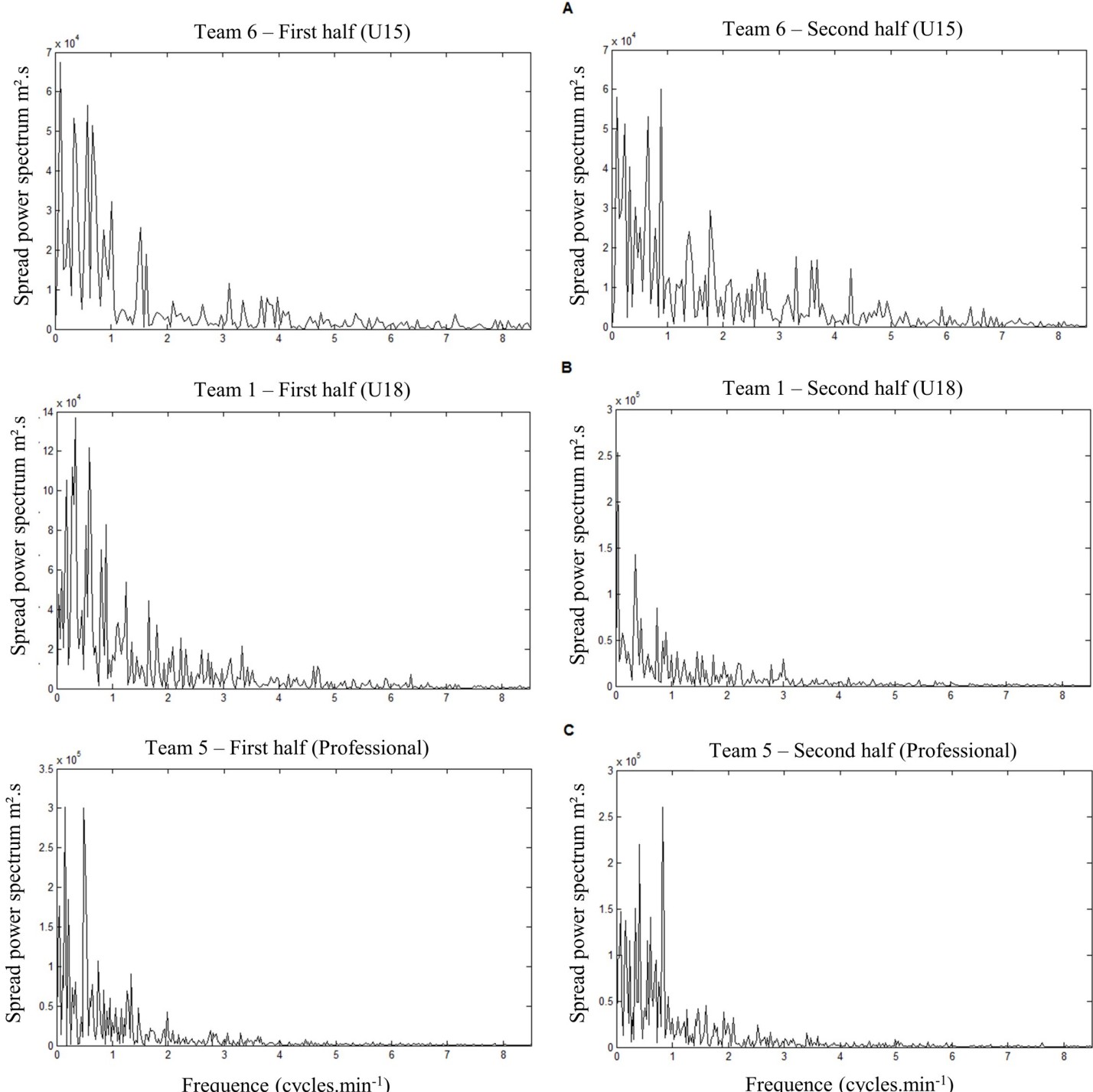

**Fig 2.** Examples power spectrum of spread as a function of frequency during the 1st and 2nd halves of play for the U15 (A), U18 (B), and professional (C) categories.

categories ($F(2,60) = 28.1$, $p < 0.01$, $ES = 1.50$). When we analyzed all values from the first to the second half, the tests did not demonstrate statistical differences ($F(1,60) = 0.23$; $p = 0.63$, $ES = 0.21$). However, observing the interaction results, we verified that there were differences ($F(2,60) = 8.15$; $p < 0.01$, $ES = 0.74$). The post-hoc test showed that, during the matches of the

**Table 1. Median frequency values (cycles.min⁻¹) for the time series of spread during the first and second halves of each team in every match analyzed for the U15, U18, and professional categories, and percentage change.**

| Matches | Teams | Median Frequency (cycles.min⁻¹) | | | | | | | | |
|---|---|---|---|---|---|---|---|---|---|---|
| | | U15 | | | U18 | | | Professional | | |
| | | First Half | Second Half | Percentage Change (%) | First Half | Second Half | Percentage Change (%) | First Half | Second Half | Percentage Change (%) |
| Match 1 | Team 1 | 1.75 | 1.61 | -8.00 | 0.82 | 0.93 | 13.41 | 0.62 | 0.59 | -4.83 |
| | Team 2 | 1.96 | 1.98 | 1.02 | 0.88 | 1.06 | 20.45 | 0.53 | 0.64 | 20.75 |
| Match 2 | Team 3 | 1.14 | 1.44 | 26.31 | 0.77 | 0.94 | 22.07 | 0.75 | 0.41 | -45.33 |
| | Team 4 | 0.96 | 1.68 | 75.00 | 0.63 | 0.94 | 49.20 | 0.75 | 0.50 | -33.33 |
| Match 3 | Team 5 | 0.65 | 1.09 | 67.69 | 0.75 | 0.55 | -26.66 | 0.61 | 0.75 | 22.95 |
| | Team 6 | 0.51 | 1.14 | 123.52 | 0.49 | 0.55 | 12.24 | 0.51 | 0.75 | 47.05 |
| Match 4 | Team 7 | 0.91 | 0.93 | 2.19 | 0.85 | 0.70 | -17.64 | 0.67 | 0.85 | 26.86 |
| | Team 8 | 0.81 | 1.35 | 66.66 | 0.64 | 0.65 | 1.56 | 0.45 | 0.56 | 24.44 |
| Match 5 | Team 9 | 0.85 | 1.05 | 23.52 | 0.76 | 0.61 | -19.73 | 0.42 | 0.50 | 19.04 |
| | Team 10 | 0.90 | 1.70 | 88.88 | 0.47 | 0.40 | -14.89 | 0.23 | 0.48 | 108.69 |

U15 category, the median frequency values of the time series of spread increased significantly from the first to the second half (Fig 3).

Fig 4 shows the median frequencies found for the time series associated with possession of the ball in the first and second halves for each match in the respective categories; each line represents a match for the categories. The exploratory analysis enabled verification that, for the U15 category, the exchange of ball possession increased in the majority of matches analyzed, from the first to the second half in 4 matches. For the U18 category, there was a drop in the frequency of possession of the ball for the majority of matches, observed in 3 matches. In the professional category, there was also a drop in the median frequency in 3 matches. It was possible to verify a greater frequency for the youngest teams for the majority of matches.

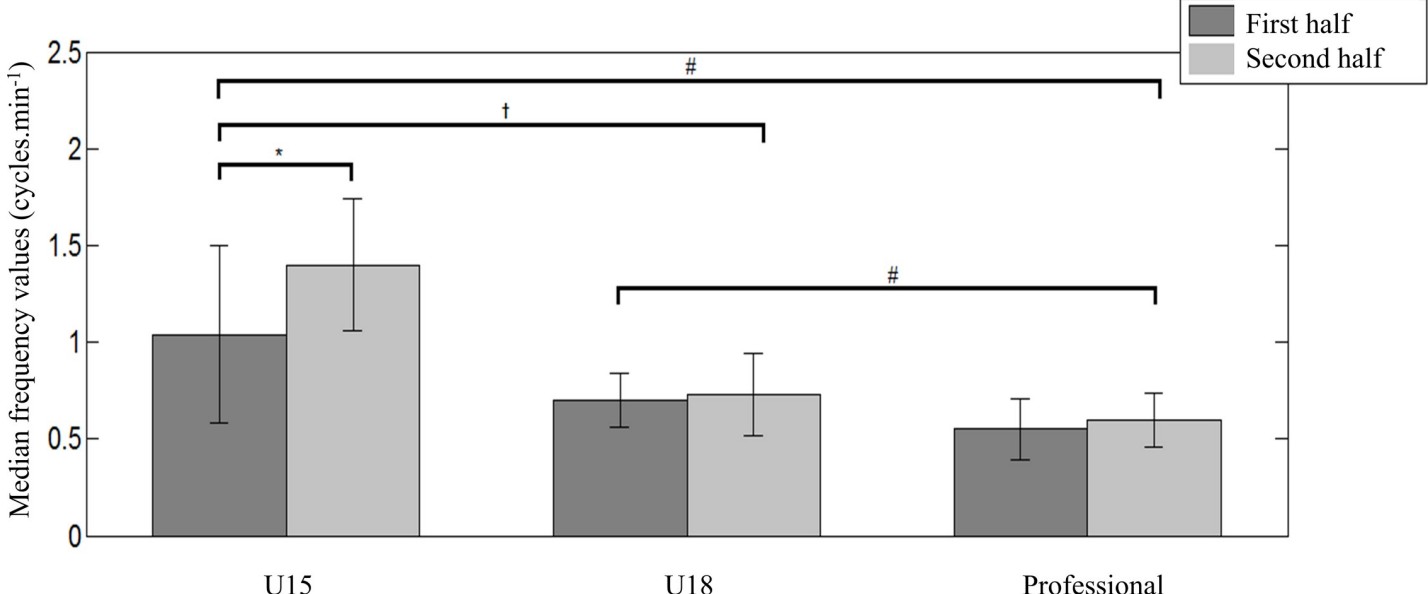

**Fig 3. Mean and standard deviation values of the median frequency of the time series of spread in futsal matches in the first and second halves for the U15, U18, and professional categories.** *$p < 0.05$; significant difference from the first to the second half, #$p < 0.05$; significantly different from the professional category, †$p < 0.05$; significantly different from the U18 category.

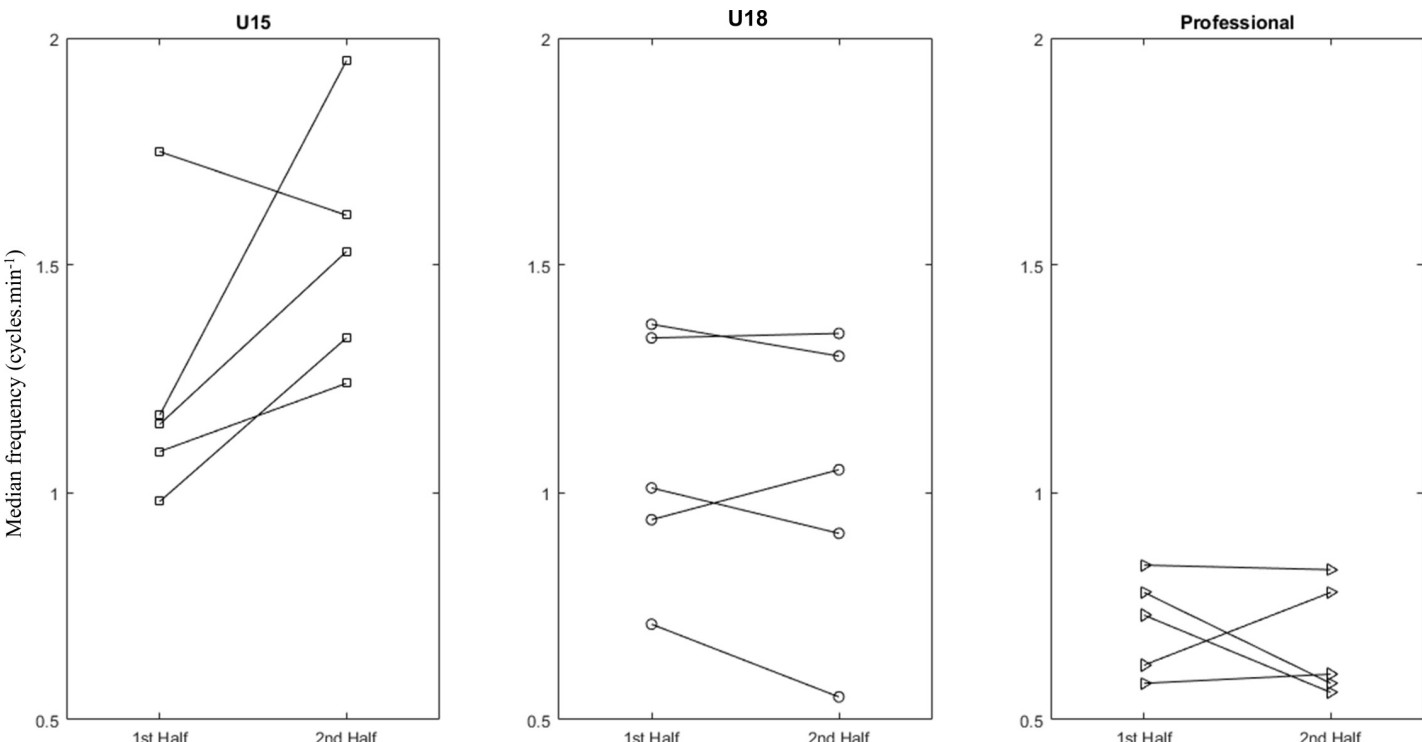

**Fig 4. Changes in the median frequency values (cycles.min$^{-1}$) for the time series of ball possession of the first and second halves of each team in every match analyzed for the U15, U18, and professional categories.**

## Discussion

The aim of the present study was to analyze the time series of team spread during futsal official matches in the frequency domain for different categories. The main results of the present study demonstrated that, in general, the spread dynamics change in futsal matches of different categories. Our first hypothesis (a) was that the younger the category, the higher the variability of the team spread would be. We confirmed this hypothesis according to the quantitative data presented in Table 1. The results indicated that the dispersion movement of the players on the court in younger categories is higher in comparison to older players, reaching values of almost two cycles.min$^{-1}$. In other words, younger teams have a tactical characteristic of spreading and compacting faster than the older categories, such as the U18 and professional. Higher frequency values for the U15 category may also be related to the magnitude of the spread values throughout the match, as seen in an example of the time series in Fig 1(A). Through visual analysis, it was possible to identify that the magnitude (i.e., the values range) of the spread values for matches in the U15 category was smaller compared to the U18 and professional categories (Fig 1(B) and 1(C), respectively). A recent study provided results that futsal players in the youngest categories present lower distribution values (surface area and spread) when compared to professionals [16]. We believe that these values could directly influence the frequency values, since if athletes spread less on the court when they are in possession of the ball, when they lose possession they compact faster, resulting in higher frequencies [5]. In another study, the authors point out that futsal is a complex sport, consider that the behavior of teams and players changes, vary, adapt on the court to obtain better organization and maintain the ball possession [22]. This may indicate that futsal is a sport that requires the learning of several

tasks, therefore, teams of younger categories must improve performance in specific tactical tasks and obtain the ball possession for a longer time.

Another explanation may be related to the time which teams remain in possession of the ball. For this study, we analyzed the median frequency of ball possession exchange for all matches. These values of ball possession frequency could help in understanding the different game dynamics [5] and, specifically for this study, in different categories (Fig 4). An exploratory analysis showed that the youngest teams presented higher values in relation to the older teams, i.e. the U15 changes ball possession between teams faster than the other categories. The U18 category was slower in relation to the U15 and faster in relation to the professional. Since this outcome may represent the speed with which players spread and compact or vice versa, this could reflect behavior changes in the transition from attack to defense and vice-versa [5], but also this dynamic can be related to higher and lower levels of experience and learning [23]. Thus, these results suggest that the dynamics of the team spread, also for futsal, could be related to the capacity of maintenance of ball possession.

The literature reports that the distances covered by players in futsal matches decrease from the first to the second period [7,9]. Therefore, the second hypothesis raised was that players would present reduced frequency values during the second half due to the decline in physical performance. From the results found in the present study, it was not possible to confirm the second hypothesis (b), since only the U15 category presented changes in the frequency values, and the values increased in the second period. There were no significant frequency changes from the first to the second period for the U18 or professional categories. Although the literature reports a decline in physical performance for professional futsal players from the first to the second half, due to the unlimited substitutions allowed by the futsal laws of the game [9], it is probable that the teams were able to maintain tactical performance throughout the match.

Particularly for the results of the U15 category, the game dynamics increased during the match, observed by the power spectrum of the first and second halves (Fig 2(A)). Again, analyzing the frequency results for the time series related to ball possession of the matches (Fig 4), values increased in the majority of matches from the first to second half. Thus, the U15 presented decreased permanence of ball possession throughout the match.

The results found in the present study could help to provide better understanding of the behavior of players' dynamics in relation to the speed with which they expand and compact during a match, reflecting the interaction between attack and defense. Coaches should be attentive to the different behavioral changes in game dynamics in each category, and adequately train their athletes who are in the transition phase so that they can adapt to the different dynamics inherent to the following categories. Furthermore, higher team spread variability associated with a higher frequency of the ball possession may reflect the team's inefficiency in maintaining ball possession, this suggests the need for coach intervention during the match or in training planning.

The limitations of the study are mainly related to data acquisition. In this study we used a video-based system to acquire player's positioning data and then calculate the analyzed variables. This method presents great accuracy and low-cost for its application [9], however, the system demands a great deal of time for image processing, which makes it difficult to collect data. Even so, as far as we know, this is the largest sample of futsal games ever reported in the literature.

## Conclusions

Based on the results of the present study we concluded that younger categories demonstrate higher spread variability than professionals, which reflects the dynamics of play between attack

and defense. However, the study repudiated the hypothesis that the speed of spreading and compacting actions would decrease from the first to the second half, as the U15 category demonstrated an increase in the game dynamics.

## Author Contributions

**Conceptualization:** Murilo José de Oliveira Bueno, Fabio Giuliano Caetano, Nicolau Melo de Souza, Sergio Augusto Cunha, Felipe Arruda Moura.

**Data curation:** Murilo José de Oliveira Bueno, Fabio Giuliano Caetano, Nicolau Melo de Souza, Sergio Augusto Cunha, Felipe Arruda Moura.

**Formal analysis:** Murilo José de Oliveira Bueno, Fabio Giuliano Caetano, Nicolau Melo de Souza, Sergio Augusto Cunha, Felipe Arruda Moura.

**Funding acquisition:** Murilo José de Oliveira Bueno, Felipe Arruda Moura.

**Investigation:** Murilo José de Oliveira Bueno, Felipe Arruda Moura.

**Methodology:** Murilo José de Oliveira Bueno, Felipe Arruda Moura.

**Project administration:** Murilo José de Oliveira Bueno, Felipe Arruda Moura.

**Supervision:** Felipe Arruda Moura.

**Writing – original draft:** Murilo José de Oliveira Bueno, Fabio Giuliano Caetano, Nicolau Melo de Souza, Sergio Augusto Cunha, Felipe Arruda Moura.

**Writing – review & editing:** Murilo José de Oliveira Bueno, Felipe Arruda Moura.

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
