## [Decision Letter · Decision Letter 0]

4 Dec 2019

PONE-D-19-25055

Variability in tactical behavior of futsal teams from different categories.

PLOS ONE

Dear Mr Bueno,

Thank you for submitting your manuscript to PLOS ONE. After careful consideration, we feel that it has merit but does not fully meet PLOS ONE’s publication criteria as it currently stands. Therefore, we invite you to submit a revised version of the manuscript that addresses the points raised during the review process.

We would appreciate receiving your revised manuscript by Jan 18 2020 11:59PM. To enhance the reproducibility of your results, we recommend that if applicable you deposit your laboratory protocols in protocols.io, where a protocol can be assigned its own identifier (DOI) such that it can be cited independently in the future. For instructions see: http://journals.plos.org/plosone/s/submission-guidelines#loc-laboratory-protocols

We look forward to receiving your revised manuscript.

Kind regards,

Filipe Manuel Clemente, PhD

Academic Editor

PLOS ONE

Journal Requirements:

3. Please state in your methods section when you conducted this study.

4. Please include in your financial disclosure statement the name of the funders of this study (as well as grant numbers if available). At present, this information is only available in your acknowledgement section.

5. Please provide additional details regarding participant consent. In the ethics statement in the Methods and online submission information, please ensure that you have specified (1) whether consent was informed and (2) what type you obtained (for instance, written or verbal, and if verbal, how it was documented and witnessed). If your study included minors, state whether you obtained consent from parents or guardians. If the need for consent was waived by the ethics committee, please include this information.

Additional Editor Comments (if provided):

Reviewers' comments:

Reviewer's Responses to Questions

**Comments to the Author**

1. Is the manuscript technically sound, and do the data support the conclusions?

Reviewer #1: Yes

Reviewer #2: Yes

2. Has the statistical analysis been performed appropriately and rigorously? 

Reviewer #1: Yes

Reviewer #2: Yes

3. Have the authors made all data underlying the findings in their manuscript fully available?

Reviewer #1: Yes

Reviewer #2: Yes

4. Is the manuscript presented in an intelligible fashion and written in standard English?

Reviewer #1: Yes

Reviewer #2: No

5. Review Comments to the Author

Reviewer #1: The main idea of the manuscript is interesting and add to previous research. However, we consdier that the paper should be improved regarding the theoretical approach that sustains the main idea, the methods and the variables used. It was not clear the link between concepts such as tactical behaviour, variability, team spread, players' age and categories.

Please add a first general paragraph introducing the futsal as a team sport that requires tactical behaviours. Also there is a need to improve the link between tactical behaviour and variability.

L55 - distribution of players in the field?

L76-78 - The issue of variability in differnt players age should be improved. There are previous studies that reported variability as a measure of adaptability while other reported variability as a measure of low proficiency. The authors should improve the explanation and clearly define and justify their position.

L78-81 - players' age and players' experience are similar concepts?

L95-96 - Please describe each group in analysis instead of the category and the championship separatedly.

L120 - Why the authors used the described method to evaluate the pread of players in the court instead of convhull, strectch index or area of play.

L157-158 - Add during the text upper or lower pannel according to the description.

Reagrding the spectral analysis teh authors could choose to present the mean results on the text and maintain the figure or the opposite. It is not necessary to present the figure and the table. The information is redundant.

Add the values of effect sizes between the 1st and 2nd half of the match in Table 1.

It is not clear how the ball possession time-series was compared with the values of frequency of spread.

A theoretical justification is required to improve the discussion of results. The authors only described the results and add some perspectives without a clear discusssion based on scientific evidences. Please improve it.

Also, there is a need to present at the end of the manuscript the practical implications of the results for match analysis and training. The results means that worst teams tend to present high levels of variability in tspread than the best ones? What this imply?

Reviewer #2: The present study aimed to analyze the time series of tactical behavior of futsal players from different categories, in the frequency domain, using an automatic tracking method. The topic is within the scope of the journal and deals with a relevant topic. The authors arrive at the conclusion that younger categories demonstrate higher tactical variability than professionals, which reflects the dynamics of play between attack and defense. Although the paper is already in a good shape, I have some concerns with regard to this paper, which need to be appropriately addressed prior to publication of this manuscript in its present form in Plos One.

Major comments

- In my opinion, the main concern refers to the real meaning of the dependent variable analyzed (i.e., team spread). Does this variable really represent the tactical behavior of the team? I think that tactical behavior is broader than simply team spread. Perhaps the team spread is a "portion" of tactical behavior. Also, does higher spread dynamics really represent greater tactical variability? This term (i.e., tactical variability) is too broad. Finally, is higher dynamic spread good or bad to players performance? It is confusing what the practical significance of this measure really is.

- It may be interesting in this study to compare the spread dynamics according to the match status in the different categories (i.e., losing vs. drawing vs. winning). This would possibly help the auhors to better explain their results, rather than just characterizing the behavior of these variables across categories.

Specific comments

- I am not a English native speaker, but I suggest a language correction.

- The aims of "abstract", introduction "and" discussion "are not in line. In my opinion, the best aim of this study is exposed in the discussion: "The aim of the present study was to analyze the time series of team spread during futsal official matches in the frequency domain for different categories.”.

- I suggest separating the “Materials and Methods” section information into subtopics (e.g., participants and match sample, design, data collection, statistical analysis).

- Please, describe in more details the contextual factors of the matches analysed (e.g., location, quality of opponents, outcome/status, team formation, playing time – see previous recommendation). These variables may influence your findings.

- Maybe match status explains the results from comparing first vs. second period.

- Figures are fuzzies and will need to be improved/changed during the review process.

- Why the Figure 1 shows only 10 minutes of the matches?

6. PLOS authors have the option to publish the peer review history of their article (what does this mean?). If published, this will include your full peer review and any attached files.

Reviewer #1: No

Reviewer #2: No

---

## [Author Response · Author response to Decision Letter 0]

22 Jan 2020

Responses to the reviewers

After incorporating the recommendations of the Academic Editor, we are resubmitting the paper “Variability in tactical behavior of futsal teams from different categories” for evaluation. We have addressed all the points raised in the review and are thankful for the editor and reviewers’ suggestions, which contributed positively to the quality of the study.

As recommended by the Academic Editor, the manuscript has been completely reformatted according to the PLOS ONE templates. We tracked all the changes in our manuscript and have added point-by-point responses to the reviewers’ comments as follows:

Academic Editor:

a) Please ensure that your manuscript meets PLOS ONE's style requirements, including those for file naming. The PLOS ONE style templates can be found at http://www.journals.plos.org/plosone/s/file?id=wjVg/PLOSOne_formatting_sample_main_body.pdf and http://www.journals.plos.org/plosone/s/file?id=ba62/PLOSOne_formatting_sample_title_authors_affiliations.pdf

Thanks for the comments. We checked all the text and corrected if necessary.

b) We note that you have stated that you will provide repository information for your data at acceptance. Should your manuscript be accepted for publication, we will hold it until you provide the relevant accession numbers or DOIs necessary to access your data. If you wish to make changes to your Data Availability statement, please describe these changes in your cover letter and we will update your Data Availability statement to reflect the information you provide.

Dear editor, we added the data to the repository with the following DOI: http://doi.org/10.6084/m9.figshare.9772559.v1

c) Please state in your methods section when you conducted this study.

Thanks for the suggestion. We added in the corresponding session the period of the study.

d) Please include in your financial disclosure statement the name of the funders of this study (as well as grant numbers if available). At present, this information is only available in your acknowledgement section.

This information was added in the last session of “Manuscript Data” on the website for resubmission

e) Please provide additional details regarding participant consent. In the ethics statement in the Methods and online submission information, please ensure that you have specified (1) whether consent was informed and (2) what type you obtained (for instance, written or verbal, and if verbal, how it was documented and witnessed). If your study included minors, state whether you obtained consent from parents or guardians. If the need for consent was waived by the ethics committee, please include this information.

Thank you for this concern. The consent form was waived by the ethics committee, this information is added to the materials and methods session.

f) PLOS requires an ORCID iD for the corresponding author in Editorial Manager on papers submitted after December 6th, 2016. Please ensure that you have an ORCID iD and that it is validated in Editorial Manager. To do this, go to ‘Update my Information’ (in the upper left-hand corner of the main menu), and click on the Fetch/Validate link next to the ORCID field. This will take you to the ORCID site and allow you to create a new iD or authenticate a pre-existing iD in Editorial Manager. Please see the following video for instructions on linking an ORCID iD to your Editorial Manager account: https://www.youtube.com/watch?v=_xcclfuvtxQ

This information has been added.

Reviewer #1:

a) The main idea of the manuscript is interesting and add to previous research. However, we consdier that the paper should be improved regarding the theoretical approach that sustains the main idea, the methods and the variables used. It was not clear the link between concepts such as tactical behaviour, variability, team spread, players' age and categories.

We appreciate the reviewer's comments and suggestions. We have addressed all the points raised by the reviewer, as follows, in point-by-point responses.

b) Please add a first general paragraph introducing the futsal as a team sport that requires tactical behaviours. Also there is a need to improve the link between tactical behaviour and variability.

The suggestion was adopted, and we reformulated the first paragraph. The link between tactical behavior and variability has also been improved.

c) L55 - distribution of players in the field?

This sentence has been reformulated.

d) L76-78 - The issue of variability in differnt players age should be improved. There are previous studies that reported variability as a measure of adaptability while other reported variability as a measure of low proficiency. The authors should improve the explanation and clearly define and justify their position.

We agree with you and the paragraph was reformulated according to suggestion.

e) L78-81 - players' age and players' experience are similar concepts?

Thank you for this concern. The sentence was reformulated.

f) L95-96 - Please describe each group in analysis instead of the category and the championship separatedly.

The correction was performed.

g) L120 - Why the authors used the described method to evaluate the pread of players in the court instead of convhull, strectch index or area of play.

We use the team spread variable for tactical analysis because it has greater robustness. It considers the distance of each player to all his teammates, so the positioning of a player has less influence on the metric compared to the convhull or strectch index.

h) L157-158 - Add during the text upper or lower pannel according to the description.

The organization of the description and figures was reformulated.

i) Reagrding the spectral analysis teh authors could choose to present the mean results on the text and maintain the figure or the opposite. It is not necessary to present the figure and the table. The information is redundant.

Dear reviewer, the results presented in Table 1 correspond to the median frequency values for each team and the corresponding halves, separately, as well as the percentage change between periods for each team. The fig. 3 presents the mean and standard deviation of the results found in the study in relation to the median frequency values of the spread of the futsal teams, as well as the results of statistical differences between the periods and categories. Fig. 4 presents individual values of median frequency results for possession ball of all teams in the first and second halves.

j) Add the values of effect sizes between the 1st and 2nd half of the match in Table 1.

Dear reviewer, the results presented in Table 1 correspond to the median frequency values for each team and the corresponding halves, separately, as well as the percentage change between periods for each team. The effect size values are presented in the text when we applied statistical tests for comparisons between the halves or categories.

k) It is not clear how the ball possession time-series was compared with the values of frequency of spread.

We analyze the variability of the ball possession as a complement to the understanding of the game dynamics. A greater variability in team spread can be explained by higher frequencies of the ball possession change, but no comparisons were made between them.

l) A theoretical justification is required to improve the discussion of results. The authors only described the results and add some perspectives without a clear discusssion based on scientific evidences. Please improve it.

Thank you for this suggestion. We improved the discussion section based on theoretical foundations.

m) Also, there is a need to present at the end of the manuscript the practical implications of the results for match analysis and training. The results means that worst teams tend to present high levels of variability in tspread than the best ones? What this imply?

Thank you, this suggestion was adopted, and practical implications were inserted in the discussion section.

Reviewer #2:

a) The present study aimed to analyze the time series of tactical behavior of futsal players from different categories, in the frequency domain, using an automatic tracking method. The topic is within the scope of the journal and deals with a relevant topic. The authors arrive at the conclusion that younger categories demonstrate higher tactical variability than professionals, which reflects the dynamics of play between attack and defense. Although the paper is already in a good shape, I have some concerns with regard to this paper, which need to be appropriately addressed prior to publication of this manuscript in its present form in Plos One.

We appreciate the reviewer's comments and suggestions. We have addressed all the points raised by the reviewer, as follows, in point-by-point responses.

Major comments

b) In my opinion, the main concern refers to the real meaning of the dependent variable analyzed (i.e., team spread). Does this variable really represent the tactical behavior of the team? I think that tactical behavior is broader than simply team spread. Perhaps the team spread is a "portion" of tactical behavior. Also, does higher spread dynamics really represent greater tactical variability? This term (i.e., tactical variability) is too broad. Finally, is higher dynamic spread good or bad to players performance? It is confusing what the practical significance of this measure really is.

Thank you for this concern. We agree with you that the team spread variable does not represent the entire tactical behavior, but it provides information about him. The terminology has been reformulated throughout the text. This term was also changed in the title. We cannot establish that the greatest variability of team spread is bad or good, only when it is associated with greater variability of ball possession.

c) It may be interesting in this study to compare the spread dynamics according to the match status in the different categories (i.e., losing vs. drawing vs. winning). This would possibly help the auhors to better explain their results, rather than just characterizing the behavior of these variables across categories.

Thank you for your comments, even the acquisition system has a high accuracy to determine the position of the players, the system demands a great deal of time for image processing. This imply a low number of games to be analyzed which was not favorable to obtain an adequate number of information under the conditions losing vs. drawing vs. winning. We believe that in future studies this methodology can be applied and thus a better understanding of the modality.

Specific comments:

d) I am not a English native speaker, but I suggest a language correction.

The mistakes have been corrected. The manuscript was submitted to a native English review. Please find attached the certificate.

e) The aims of "abstract", introduction "and" discussion "are not in line. In my opinion, the best aim of this study is exposed in the discussion: "The aim of the present study was to analyze the time series of team spread during futsal official matches in the frequency domain for different categories.”.

Thank you for this concern. We corrected the aim of the study in the abstract and introduction sections.

f) I suggest separating the “Materials and Methods” section information into subtopics (e.g., participants and match sample, design, data collection, statistical analysis).

We adopted your suggestion and the materials and methods section was reformulated.

g) Please, describe in more details the contextual factors of the matches analysed (e.g., location, quality of opponents, outcome/status, team formation, playing time – see previous recommendation). These variables may influence your findings.

We didn't get some information, which may be one of the limitations of the studies because we don't have access to the player and coaches. We only had access with the coordinators and directors of the championships to obtain an authorization for the filming.

h) Maybe match status explains the results from comparing first vs. second period.

Thank you for your comments, even the acquisition system has a high accuracy to determine the position of the players, the system demands a great deal of time for image processing. This imply a low number of games to be analyzed. With a greater number of games, we believe that in future studies this methodology can be applied for a better understanding of the related to match results.

i) Figures are fuzzies and will need to be improved/changed during the review process.

The fig. 1 presents examples of the time series of spread for 10 min of first halve in the U15, U18 and Professional categories. The fig. 2, exemplifies the spectral analyzes for the time series of spread, being presented for a team of each category in the first and second halve. The fig. 3 presents the mean and standard deviation of the results found in the study in relation to the median frequency values of the spread of the futsal teams, as well as the results of statistical differences between the periods and categories. Fig. 4 presents individual values of median frequency results for possession ball of all teams in the first and second halves.

If you have any questions about the figures, please, let us know that we will certainly change them to the final version.

j) Why the Figure 1 shows only 10 minutes of the matches?

We chose to represent only 10 minutes of the matches to improve the visualization of the oscillations on the spread time series for all categories. Again, if you have any questions about the figures, please, let us know that we will certainly change them to the final version.

We would like to thank the reviewers and editor for their contributions and suggestions, which contributed positively to the quality of the study. We hope that the revision of our manuscript meets the publication standards of PLOS ONE.

The authors.

---

## [Decision Letter · Decision Letter 1]

27 Feb 2020

PONE-D-19-25055R1

Variability in tactical behavior of futsal teams from different categories.

PLOS ONE

Dear Mr Bueno,

Thank you for submitting your manuscript to PLOS ONE. After careful consideration, we feel that it has merit but does not fully meet PLOS ONE’s publication criteria as it currently stands. Therefore, we invite you to submit a revised version of the manuscript that addresses the points raised during the review process.

Please consider the final minor corrections recommended by the reviewers

We would appreciate receiving your revised manuscript by Apr 12 2020 11:59PM. To enhance the reproducibility of your results, we recommend that if applicable you deposit your laboratory protocols in protocols.io, where a protocol can be assigned its own identifier (DOI) such that it can be cited independently in the future. For instructions see: http://journals.plos.org/plosone/s/submission-guidelines#loc-laboratory-protocols

We look forward to receiving your revised manuscript.

Kind regards,

Filipe Manuel Clemente, PhD

Academic Editor

PLOS ONE

Reviewers' comments:

Reviewer's Responses to Questions

**Comments to the Author**

1. If the authors have adequately addressed your comments raised in a previous round of review and you feel that this manuscript is now acceptable for publication, you may indicate that here to bypass the “Comments to the Author” section, enter your conflict of interest statement in the “Confidential to Editor” section, and submit your "Accept" recommendation.

Reviewer #2: All comments have been addressed

Reviewer #3: All comments have been addressed

2. Is the manuscript technically sound, and do the data support the conclusions?

Reviewer #2: Yes

Reviewer #3: Yes

3. Has the statistical analysis been performed appropriately and rigorously? 

Reviewer #2: Yes

Reviewer #3: Yes

4. Have the authors made all data underlying the findings in their manuscript fully available?

Reviewer #2: Yes

Reviewer #3: Yes

5. Is the manuscript presented in an intelligible fashion and written in standard English?

Reviewer #2: Yes

Reviewer #3: Yes

6. Review Comments to the Author

Reviewer #2: All comments of the Academic Editor and Reviewers have been addressed. Therefore, I accept the manuscript in the current form.

Reviewer #3: The article “Variability in tactical behavior of futsal teams from different categories” has a very relevant theme and I consider it to be appropriate to the scope of the PLos One. The variables of tactical behaviors in Futsal need further studies.

A suggestion for better clarity of the work and possible discussions about the athletes' tactical behaviors, would it be to put more information about the registered games, for example, were they games of the qualifying or final phase championship? Was it necessary to win to continue in the championship? We have observed that the tactical behavior of some athletes changes according to the difficulty or value of the game played. I think it would be interesting to make this information about games clear at work, even for new questions in the future.

In my opinion, after making such additions to the information in the description of the sample used, the work will be ready for publication, since the placements of the other reviewers were relevant and met by the vast majority of authors. I agree with all the other reviewers' comments, and since they were reviewed by the authors, I will not post again.

Congratulation for their work!

7. PLOS authors have the option to publish the peer review history of their article (what does this mean?). If published, this will include your full peer review and any attached files.

Reviewer #2: Yes: Rodrigo Aquino

Reviewer #3: No

---

## [Author Response · Author response to Decision Letter 1]

28 Feb 2020

Responses to the reviewers

After incorporating the recommendations of the Academic Editor, we are resubmitting the paper “Variability in tactical behavior of futsal teams from different categories” for evaluation. We have addressed all the points raised in the review and are thankful for the editor and reviewers’ suggestions, which contributed positively to the quality of the study.

We tracked all the changes in our manuscript and have added point-by-point responses to the reviewers’ comments as follows:

Reviewer #2:

a) All comments of the Academic Editor and Reviewers have been addressed. Therefore, I accept the manuscript in the current form.

Thank you for the reviewer’s suggestions, which contributed positively to the quality of the study.

Reviewer #3:

a) The article “Variability in tactical behavior of futsal teams from different categories” has a very relevant theme and I consider it to be appropriate to the scope of the Plos One. The variables of tactical behaviors in Futsal need further studies. 

We appreciate the reviewer's comments and suggestions. We have addressed all the points raised by the reviewer, as follows, in point-by-point responses.

b) A suggestion for better clarity of the work and possible discussions about the athletes' tactical behaviors, would it be to put more information about the registered games, for example, were they games of the qualifying or final phase championship? Was it necessary to win to continue in the championship? We have observed that the tactical behavior of some athletes changes according to the difficulty or value of the game played. I think it would be interesting to make this information about games clear at work, even for new questions in the future.

Thank you for this suggestion. We added detailed information to the corresponding paragraph.

We would like to thank the reviewers and editor for their contributions and suggestions, which contributed positively to the quality of the study. We hope that the revision of our manuscript meets the publication standards of PLOS ONE.

The authors.

---

## [Editor Report · Decision Letter 2]

3 Mar 2020

Variability in tactical behavior of futsal teams from different categories.

PONE-D-19-25055R2

Dear Dr. Bueno,

We are pleased to inform you that your manuscript has been judged scientifically suitable for publication and will be formally accepted for publication once it complies with all outstanding technical requirements.

With kind regards,

Filipe Manuel Clemente, PhD

Academic Editor

PLOS ONE
---

## [Editor Report · Acceptance letter]

5 Mar 2020

PONE-D-19-25055R2 

Variability in tactical behavior of futsal teams from different categories. 

Dear Dr. Bueno:

I am pleased to inform you that your manuscript has been deemed suitable for publication in PLOS ONE. Congratulations! Your manuscript is now with our production department. 

With kind regards,

on behalf of

Dr. Filipe Manuel Clemente 

Academic Editor

PLOS ONE